# Revisiting Noise Resilience Strategies in Gesture Recognition: Short-Term Enhancement in sEMG Analysis

## Abstract

Gesture recognition based on surface electromyography (sEMG) has been gaining importance in many 3D Interactive Scenes. However, sEMG is easily influenced by various forms of noise in real-world environments, leading to challenges in providing long-term stable interactions through sEMG. Existing methods often struggle to enhance model noise resilience through various predefined data augmentation techniques. In this work, we revisit the problem from a short-term enhancement perspective to improve precision and robustness against various common noisy scenarios with learnable denoise using sEMG intrinsic pattern information and sliding-window attention. We propose a Short Term Enhancement Module(STEM), which can be easily integrated with various models. STEM offers several benefits: 1) Noise-resistant, enhanced robustness against noise without manual data augmentation; 2) Adaptability, adaptable to various models; and 3) Inference efficiency, achieving short-term enhancement through minimal weight-sharing in an efficient attention mechanism. In particular, we incorporate STEM into a transformer, creating the Short-Term Enhanced Transformer (STET). Compared with best-competing approaches, the impact of noise on STET is reduced by more than 20%. We report promising results on classification and regression tasks and demonstrate that STEM generalizes across different gesture recognition tasks. The code is available at `https://anonymous.4open.science/r/short_term_semg`.

## 1 Introduction

Surface Electromyographic (sEMG) is a non-invasive technique for monitoring muscle neurons firing, which is an effective way to capture human motion intention and has shown great application potential in the field of human-computer interaction (HCI) Xiong et al. (2021); Liu et al. (2021b; 2020). A schematic diagram of the EMG-based HCI System is shown in Figure 1. Compared to traditional HCI channels, sEMG has the advantages of being generated prior to actual motion (50-150 ms), containing rich motion intention information, and being easy to collect Sun et al. (2020). Therefore, there has been increasing interest in exploring EMG-based motion track Liu et al. (2021a) and pathological analysis.

By treating sEMG as time series, deep sequential models Bi et al. (2019); Tsinganos et al. (2019); Becker et al. (2018); Li et al. (2021); Du et al. (2017) have been applied to sEMG modeling. For example, Zhang *et al.* employs a multi-task encoder-decoder framework improve the robustness of sEMG-based Sign Language Translation (SLT) Zhang et al. (2022). Rahimian *et al.* employs Vision Transformer (ViT)-based architecture (TEMGNet) to improve the accuracy of sEMG-based myocontrol of prosthetics Rahimian et al. (2021). Although these methods demonstrate enhanced performance compared to traditional approaches, they process sEMG signals as generic time-series data without specifically tailoring their design to address the unique characteristics of sEMG, such as their high variability and sensitivity to external noise and interference. This oversight leads to issues such as difficulty in handling low signal-to-noise ratios due to changes in skin surface conditions and signal interference, and failing to capture subtle but important motion information in sEMG signals. As a result, the robustness and accuracy of existing models remain significantly challenged.

Processing sEMG signals is challenging due to the complex noise mixed in the skin's surface and the presence of patterns across various time scales. Existing works, mainly focusing on long-term sequences, have used transformers to treat sEMG as a typical time series, aiming to enhance long-term dependencies. These

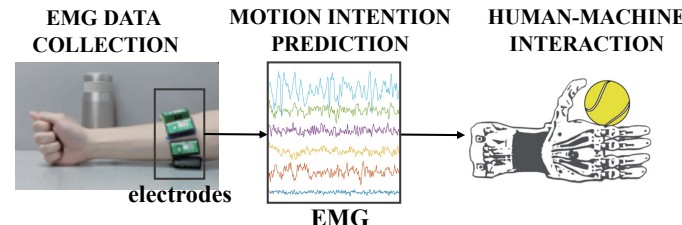

Figure 1: Schematic Diagram of EMG-based Human-Computer Interaction System.

approaches overlook the critical features which present in short-time scales. Short-time scale features are important in sEMG analysis, as they aid in distinguishing subtle movements and facilitate the removal of variable noise. For example, gestures like Index Finger Extension (IFE) and Middle Extension (ME), while similar in global sEMG patterns, can be differentiated through localized short-term signal variations.

To this end, in this paper, we present a lightweight but powerful Module called Short-Term Enhanced Module (STEM) which utilizes sliding window attention with weight sharing to capture short-term features. Building on STEM, we further propose the Short-Term Enhanced Transformer (STET). STET leverages STEM to capture local signal changes, enhancing noise resistance, and then combines STEM with long-term features, further improving accuracy for downstream predictions. Furthermore, to enhance model robustness with minimal annotation, we propose a self-supervised paradigm based on sEMG Signal Masking to leverage the inherent variability in sEMG signals.

Finally, we conducted extensive experiments on the largest public sEMG datasets. The experimental results show that STET surpasses existing methods by a significant margin in both gesture classification and joint angle regression tasks for single-finger, multi-finger, wrist, and rest gestures. Meanwhile, STET achieves strong robustness even when trained on pure data and tested on noisy data. Compared with best-competing approaches, the impact of noise on STET is reduced by more than 20%. Moreover, through visualizations, we show that the long-term and short-term features are complementary in sEMG-based gesture recognition tasks, and the fusion of the two features can make the classification boundary more obvious. This clearly demonstrates that short-term information is critical for sEMG-based gesture recognition and will provide a new design paradigm for future sEMG model design. In particular, we have deployed STET as an important functional component in our HCI system, which can offer a more intuitive and effective experience. Our real-world deployment is shown in the appendix 5.

To the best of our knowledge, we are the first to highlight the short-term features in sEMG-based gesture recognition. Our contributions can be summarized as follows:

1) From the perspective of enhancing short-term features, we propose STEM, a learnable and adaptable, noise-resistant module. The integration of STEM into various neural networks has resulted in a marked improvement in their performance;

2) we introduce sEMG Signal Masking to self-supervised sEMG Intrinsic Pattern Capture Module to leverage the inherent variability in sEMG;

3) we conduct experiments on the largest wrist sEMG dataset, showing that our proposed method outperforms existing approaches in terms of accuracy and robustness. Short-term enhancement can be extended to other models like Informer.

## 2 RELATED WORK

**The EMG-based Intention Prediction of Human Motion** can be broadly divided into model-based and data-driven methods. Model-based methods typically combine disciplines such as kinesiology, biomechanics, and human dynamics to explicitly model the relationship between EMG and outputs (such as joint angles and forces). The model often includes specific parameters, such as joint positions and bone-on-bone friction, that need to be repeatedly experimented with and adjusted until the desired performance is achieved. In terms of parameter selection and determination, model-based methods can be further divided into kinematic models Borbély & Szolgay (2017), dynamic models Koike & Kawato (1995); Koirala et al. (2015); Liu et al. (2015), and muscle-bone models Wang & Buchanan (2002); Zhao et al. (2020); Yao et al. (2018). Clancy *et al.* used a nonlinear dynamics model to identify the relationship between constant posture electromyography and torque at the elbow joint Clancy et al. (2012). Hashemi *et al.* used the Parallel Cascade Identification method to

establish a mapping between forearm muscles and wrist forces Hashemi et al. (2012). However, model-based methods have a large number of parameters that are difficult to measure directly. Currently, only simple motion estimation with a limited number of joints and degrees of freedom is possible. In contrast to model-based approaches, data-driven methods do not require the measurement of various parameters. Recently, some researchers have begun to use temporal deep learning models to extract motion information from sEMG Lin et al. (2022); Zhang et al. (2022); Guo et al. (2021). Lin *et al.* proposed a BERT-based structure to predict hand movement from the Root Mean Square (RMS) feature of the sEMG signal Lin et al. (2022). Rahimian *et al.* proposed a novel Vision Transformer (ViT)-based neural network architecture to classify and recognize upper-limb hand gestures from sEMG for use in myocontrol of prostheses Rahimian et al. (2021). However, these methods have neglected the modeling of short-term dependencies and have not considered the inherent variability in sEMG signals.

## 3 PRELIMINARIES

### 3.1 DATASET

We conduct the experiments on the GRABMyo Dataset Pradhan et al. (2022) and the Ninapro DB2 Atzori et al. (2014), which are the largest and most widely used sEMG datasets and have great potential for developing new generation human-machine interaction based on sEMG.

*Data processing.* The subjects performed 17 gestures of hand and wrist (including a rest period sEMG) according to the prompts on the computer screen. Each gesture was repeated 7 times, each lasting 5 seconds. In order to improve the convergence speed of the model, we use two methods (Max-Min normalization, $\mu$-law normalization) to normalize the data Rahimian et al. (2020); Recommendation (1988). After normalization, we use a time-sliding window to split samples. We set the window size as 200ms, and the overlap of adjacent windows is 10ms. $\mu$-law normalization can logarithmically amplify the outputs of sensors with small magnitudes, which results in better performance than linear normalization.

**Definition 1 (sEMG Signal Sequence)** *An sEMG signal sequence is defined as a temporal signal sequence sampled by multiple sensors from a human wrist, which can be formulated as* $\mathbf{X} = [\mathbf{x}_1, \mathbf{x}_2, .., \mathbf{x}_t]$, *where* $t$ *is the time window.* $\mathbf{x}_i = [x_{i,1}, x_{i,2}, ...x_{i,c}]$ *represents the signal vector of* $c$ *sensors, where* $x_{i,j}$ *is the signal value of the* $j$-*th sensor in the* $i$-*th time step.*

## 4 TECHNOLOGY DETAIL

### 4.1 MODEL OVERVIEW

Figure 2 illustrates the overview of our proposed framework for gesture recognition, which contains three components: (1) The *sEMG Intrinsic Pattern Capture* module encodes the sEMG signal sequence into the hidden sEMG representations. A pre-training model with a segment masking strategy and MSE reconstructing loss is proposed to learn inherent variability from the sEMG signals into the model's parameters. (2) The *Long-term and Short-term Enhanced* module uses two decoupling heads to extract the long-term and short-term context information separately, which improves the sEMG representations in preserving both the global sEMG structure and multiple local signal changes of the sEMG. (3) The *Asymmetric Optimization* strategy addresses the problems of sample biases and imbalance in gesture recognition via an asymmetric classification loss, which can make the model focus on hard and positive samples to improve the recognition.

### 4.2 SEMG INTRINSIC PATTERN CAPTURE MODULE

#### 4.2.1 SEMG SIGNAL ENCODING

Given the sEMG signal sequence $\mathbf{X} = [\mathbf{x}_1, \mathbf{x}_2, .., \mathbf{x}_t]$, we first project each signal $\mathbf{x}_i \in \mathbb{R}^c$ into a hidden embedding via a transformation matrix and add each signal embedding with an absolute position embedding. Then, we feed the output sequence into a $L$-layer Transformer and obtain the output signal embeddings $\mathbf{X}^{(L)} = [\mathbf{x}_1^{(L)}, \mathbf{x}_2^{(L)}, .., \mathbf{x}_t^{(L)}]$, which incorporate temporal context signal information for each position in the sequence.

#### 4.2.2 SEMG SIGNAL MASKING

After the sEMG signal-extracting module is constructed, we aim to use pre-training to exploit the intrinsic pattern and temporal semantics disclosed by the unlabeled sEMG signals (labeling sEMG is time-consuming and labor-intensive) and give a good initialization for the model parameters, then

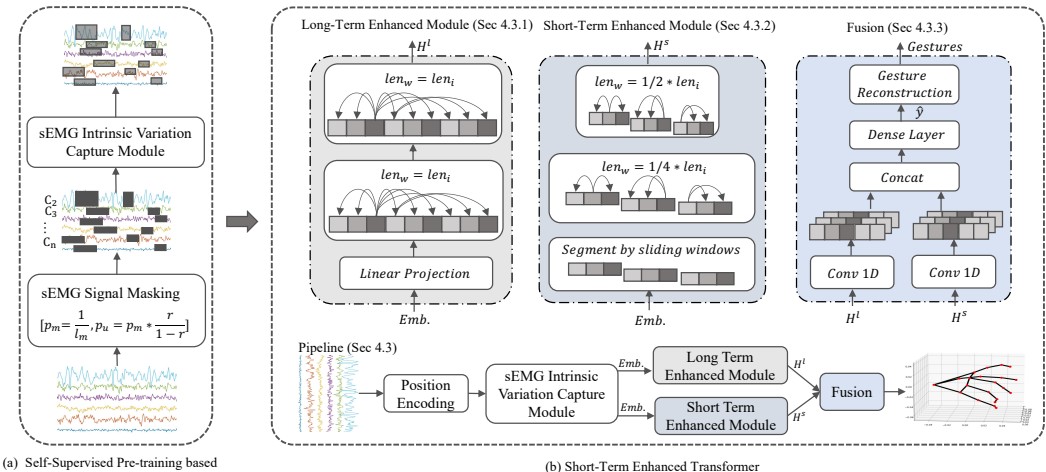

Figure 2: Overview of STET. The sEMG signal is encoded using the sEMG Intrinsic Pattern Capture module, which is first pre-trained via sEMG signal Masking. A long-term and short-term enhanced module improves sEMG representations. An asymmetric optimization strategy addresses biases and imbalances in gesture recognition through an asymmetric classification loss.

avoid the model focusing on some noisy features in the supervised learning task so as to over-fitting on some local minimums. Thus, we propose a sEMG Intrinsic Pattern Capture based on a signal masking strategy, which is detailed in appendix 1.

Specifically, given a transformed signal embedding sequence $\mathbf{X} = [\mathbf{x}_1, \mathbf{x}_2, .., \mathbf{x}_t]$, instead of adding masks on the sequence in terms of time steps like BERT, we add sensor-wise masks for the signal sequence of each sensor similar with Zerveas et al. (2021), which can encourage the model to learn more fine-grained temporal context dependency on the signal sequence of multiple electrodes. For the signal sequence of the $i$-th sensor, formulated as $[x_{1,i}, x_{2,i}, ..., x_{t,i}]$, i.e, the $i$-th column of $\mathbf{X}$, we generate a binary mask vector $\mathbf{m}_i \in \mathbb{R}^t$, where average $r$ radio of elements in $\mathbf{m}_i$ should be 0.15. Randomly generating $\mathbf{m}_i$ may cause a lot of isolated-masked signals, meaning one masked signal whose adjacent signals are unmasked. However, a single signal can be easily predicted by its immediately preceding or succeeding signals, making self-supervised learning easy to fitting on ineffective patterns and poor for learning temporal semantic information. In consideration of this, we introduce a more complex masking strategy that aims to generate multiple masked segments on the sequence with an average length $l_m$, which means $m_i$ is composed of contiguous masked segments and unmasked segments. The length of masked segments follows a geometric distribution with mean $l_m$, and the length of unmasked segments follows a geometric distribution with mean $l_u$. Also, the $\frac{l_m}{l_u} = \frac{r}{1-r}$ so that the number of masked elements would follow the proportion $r$. The pseudocode of the masking algorithm is presented in Algorithm 1.

Then, we can mask the input sEMG signal sequence $\mathbf{X}$ by $\widehat{\mathbf{X}} = \mathbf{X} \odot \mathbf{M}$, where $\odot$ is elementwise multiplication and $\widehat{\mathbf{X}}$ is the masked input. With the proposed Transformer-based sEMG signal encoder, we can obtain the output $\widehat{\mathbf{X}}^{(L)} = [\widehat{\mathbf{x}}_1^{(L)}, \widehat{\mathbf{x}}_2^{(L)}, ..., \widehat{\mathbf{x}}_t^{(L)}]$. For self-supervised learning, we add a linear layer on the top of masked output to reconstruct each sEMG signal $\widehat{\mathbf{x}}_i^{(L)}$ as $\widetilde{\mathbf{x}}_i \in \mathbb{R}^c$, which is the reconstructed sEMG signal in the $i$-th time step generated from the masked input. Then, we minimize the Mean Squared Error (MSE) of the reconstructed signals and original signals on the masked positions for each sample:

$$\min \frac{1}{|\mathbf{M}|} \sum_{i=0}^{t} \sum_{j=0}^{c} \mathbb{1}(\mathbf{M}_{i,j} = 0)(\widetilde{x}_{i,j} - x_{i,j})^2, \tag{1}$$

where $\mathbb{1}(\cdot)$ is the indicator function, $\widetilde{x}_{i,j}$ and $x_{i,j}$ are the reconstructed value and original value of $j$-th sensor in $\widetilde{\mathbf{x}}_i$ and $\mathbf{x}_i$ respectively, and $\mathbf{M}_{i,j}$ is the element in the $i$-th row and the $j$-th column of $\mathbf{M}$. Thus, we can pre-train the sEMG Intrinsic Pattern Capture via the above strategy to obtain well-initialized model parameters for the downstream task. In practice, we empirically set the masking

proportion $r$ as 0.15 and the average length of masked segments as 3. The illustration of the pre-training procedure is in Figure 2 (a).

### 4.3 Long-term and Short-term Decoding

Then, based on the pre-trained sEMG Intrinsic Pattern Capture, we develop two decoder heads to further extract the long-term and short-term dependency on the signal sequences, respectively. Intuitively, both the long-term and short-term information on signal sequences are significant in the gesture recognition problem. Long-term information refers to the global context of an sEMG sequence, which provides the overall structure of a signal to help the interpretation of the gesture. Short-term information refers to the movement signal in a short time interval of the whole sequence, which can provide specific local characteristics for accurate recognition when the overall structures of sEMGs are ambiguous. For example, distinguishing between Index Finger Extension (IFE) and Middle Extension (ME) movements requires a closer examination of the local signal changes in sEMG, whereas differentiating gestures with large variations, such as hand gestures and wrist gestures, necessitates a focus on the global sEMG information.

#### 4.3.1 Preserving Long-term sEMG Signal

Given the hidden output $\mathbf{X}^{(L)} = [\mathbf{x}_1^{(L)}, \mathbf{x}_2^{(L)}, ..., \mathbf{x}_t^{(L)}]$ of a sEMG signal sequence, we first build a long-term decoder to extract the long-term dependency on the complete output. Specifically, the long-term decoder is defined as a multi-head self-attention layer.

Through the self-attention layer, the global context signal information is collected to the embeddings of $t$ timesteps with different attention weights. The detailed equation is shown in appendix.

#### 4.3.2 Preserving Short-term sEMG Signal

To model the local context information within a short time interval, we introduce a slide-window self-attention layer to extract the short-term dependency on the signal outputs. Similarly, we stack multiple attention heads and calculate the attention of context signals to weighted sum them up into the final representations. The difference is that, for each time step, we only calculate the attention of its nearest $w$ context. Specifically, we can rewrite the $\text{Attention}(\cdot)$ in Eq.4b as:

$$\text{Attention}_S(\mathbf{Q}, \mathbf{K}, \mathbf{V}) = \left[ \text{Softmax}\left( \frac{\mathbf{Q}_i \mathbf{K}_i^{wT}}{\sqrt{h}} \right) \mathbf{V}_i^w \right]_{i=1}^t, \tag{2a}$$

$$[\mathbf{K}_i^w]_{i=1}^t = \text{Unfold}(\mathbf{K}, w), \tag{2b}$$

$$[\mathbf{V}_i^w]_{i=1}^t = \text{Unfold}(\mathbf{V}, w), \tag{2c}$$

where $w$ is the sliding windows size, $\mathbf{Q}_i \in \mathbb{R}^h$ as the $i$-th query is the $i$-th row of $\mathbf{Q}$, $\mathbf{K}_i^w \in \mathbb{R}^{w \times h}$ and $\mathbf{V}_i^w \in \mathbb{R}^{w \times h}$ are the keys and values in a window around the $i$-th query. For the key and value matrix, we utilize the $Unfold(\cdot)$ operation to generate the sliding windows for each timestep. Noted that to avoid confusion, we omit the index of attention head in the above equation.

Thus, by stacking multiple sets of parameters in $\text{Attention}_S(\cdot)$ to constitute different attention heads, we can obtain the short-term sEMG embeddings $\mathbf{H}^s \in \mathbb{R}^{t \times h}$ by $\mathbf{H}^s = \text{MultiHead}_S(\mathbf{X}^{(L)})$. Using sliding windows, each row in $\mathbf{H}^s$ preserves the local context information of the corresponding timestep, representing the movement from the past $w/2$ timesteps to the next $w/2$ timesteps.

Unlike Focal Transformer Yang et al. (2021), which process Long Term Feature and Short Term Feature sequentially, we handle them in parallel. Unlike LST-EMG-Net, which employs a patch-based segmentation approach, our method leverages a sliding-window attention mechanism, enabling finer-grained feature extraction and better sensitivity to signal variations across boundaries.

#### 4.3.3 Fusion

Obtained the long-term embeddings $\mathbf{H}^l$ and the short-term embeddings $\mathbf{H}^s$ of an sEMG signal sequence, we first concatenate them in terms of the hidden dimension, then introduce a 1-D convolution to summarize the $t$-step sEMG embedding sequence into the final sEMG representation, which is fed into a *Feed Forward Layer* with a *Sigmoid Layer* to obtain the final classification probability

of which gesture the sEMG belonging to, which can be written as: $\widehat{\mathbf{y}} = \sigma(\text{FC}(\mathbf{u}^T \cdot [\mathbf{H}^l : \mathbf{H}^s]))$, where $\text{FC}(\cdot)$ is a two-layer fully connected layer, $\sigma(\cdot)$ is the activation function, and $\widehat{\mathbf{y}} \in \mathbb{R}^C$ is the output classification probability of the sEMG signals.

## 4.4 ASYMMETRIC OPTIMIZATION

As the common use in multi-label classification, we reduce the gesture recognition problem into a series of binary classification tasks. However, we consider two tricky problems that exist in the above model optimization. (1) As the sampled signals are usually unstable over time, the samples of the sEMG signal sequence may be critically biased. Some samples with strong signals are easily predicted, while many samples with fuzzy signals are hard to predict. (2) We set 17 classes for the sEMG signals, and each class contains a comparable number of samples. Thus, each class contains, on average, many more negative samples than positive ones. This imbalance may cause the model to eliminate the gradients from the positive samples in the optimization process, resulting in poor accuracy. Realizing this, we introduce the Asymmetric loss Ridnik et al. (2021) for the gesture classification task. Asymmetric loss is a variant of Focal loss. (1) It uses focusing parameters to reduce the contribution of easily predicted samples and make the model optimization focus on hard samples; (2) It further introduces asymmetric focusing parameters and asymmetric probability shifting to down-weight the contribution from massive easy negatives and emphasizes the contribution of positive samples. Thus, we define the loss function as follows:

$$\mathcal{L}_{STET} = -\sum_{i=1}^{N}\sum_{j=1}^{C}\left(y_{i,j}\left(1 - \widehat{y}_{i,j}\right)^{\gamma^+}\log\left(\widehat{y}_{i,j}\right) + \left(1 - y_{i,j}\right)\left(\widehat{y}_{i,j}^{m}\right)^{\gamma^-}\log\left(1 - \widehat{y}_{i,j}^{m}\right)\right), \quad (3)$$

and $\widehat{y}_{i,j}^{m} = \max\left(\widehat{y}_{i,j} - m, 0\right)$, where $y_{i,j}$ and $\widehat{y}_{i,j}$ is the ground-truth and probability of the $i$-th sEMG signal sequence belonging to the $j$-th gesture. $\left(1 - \widehat{y}_{i,j}\right)^{\gamma^+}$ and $\left(\widehat{y}_{i,j}^{m}\right)^{\gamma^-}$ are two terms to make the weights of hard predicted samples bigger than those easily predicted samples, $\gamma^+$, and $\gamma^-$ are two focusing parameters and $\gamma^+ > \gamma^-$ lead to asymmetric focusing that help the optimization pay more focus on positive samples of each class. $\widehat{y}_{i,j}^{m}$ is the shifted probability and $m$ is shifting margin. The probability shifting for negative samples encourages the optimizer to further reduce their contribution.

## 5 EXPERIMENT

### 5.1 SETTINGS

**Implementation Details** STET is implemented in PyTorch Paszke et al. (2019) and is trained using one RTX 3090 GPU. During training, we use the RAdam Liu et al. (2019), which is a theoretically sound variant of the Adam optimizer with a weight decay of 1e-3. For classification tasks, we conduct user-specific pretraining on the GRABMyo dataset, and for regression tasks, we carry out user-specific pretraining on the NinaPro DB2 dataset. In the decoder, we use two layers of full attention in the long-term decoder and two layers of sliding window attention in the short-term decoder. The short-term decoder's window size is 41 and 21, and the window's move step is set to 1. In both the pre-training and fine-tuning periods, we set the batch size to 16 and set drop out to 0.2.

| Model | Single-finger | | Multi-finger | | Wrist | | Rest | | Overall | |
|---|---|---|---|---|---|---|---|---|---|---|
| | ACC | STD | ACC | STD | ACC | STD | ACC | STD | ACC | STD |
| Asif *et al.* Asif et al. (2020) | 83.44% | 0.015 | 83.58% | 0.013 | 89.40% | 0.009 | 90.86% | 0.012 | 85.34% | 0.014 |
| TCN Tsinganos et al. (2019) | 78.78% | 0.017 | 79.10% | 0.018 | 87.27% | 0.011 | 88.57% | 0.017 | 81.50% | 0.016 |
| GRU Chen et al. (2021) | 84.45% | 0.015 | 84.88% | 0.013 | 90.06% | 0.009 | 89.42% | 0.019 | 86.30% | 0.015 |
| TEMGNet Rahimian et al. (2021) | 77.70% | 0.019 | 74.00% | 0.029 | 84.04% | 0.017 | 87.46% | 0.014 | 78.02% | 0.019 |
| Zerveas *et al.* Zerveas et al. (2021) | 78.45% | 0.016 | 77.20% | 0.020 | 87.28% | 0.016 | 86.76% | 0.017 | 80.43% | 0.018 |
| Informer Zhou et al. (2021) | 86.88% | 0.016 | 86.54% | 0.017 | 91.90% | 0.011 | 83.56% | 0.024 | 87.71% | 0.016 |
| LST-EMG-Net Zhang et al. (2023) | 87.21% | 0.011 | 83.16% | 0.012 | 88.36% | 0.018 | 82.52% | 0.021 | 85.31% | 0.015 |
| TEMGNET+STEM(ours) | 84.57% | 0.017 | 81.23% | 0.022 | 88.12% | 0.017 | 88.74% | 0.013 | 84.07% | 0.017 |
| Informer+STEM(ours) | 87.42% | 0.015 | 88.39% | 0.018 | 92.07% | 0.013 | 90.33% | 0.011 | 89.14% | 0.015 |
| STET | **88.27%** | 0.014 | **89.93%** | 0.015 | **93.77%** | 0.010 | **95.33%** | 0.012 | **90.76%** | 0.012 |

Table 1: The gesture classification performance on the Single-finger, Multi-finger, Wrist, Rest, and Overall categories.

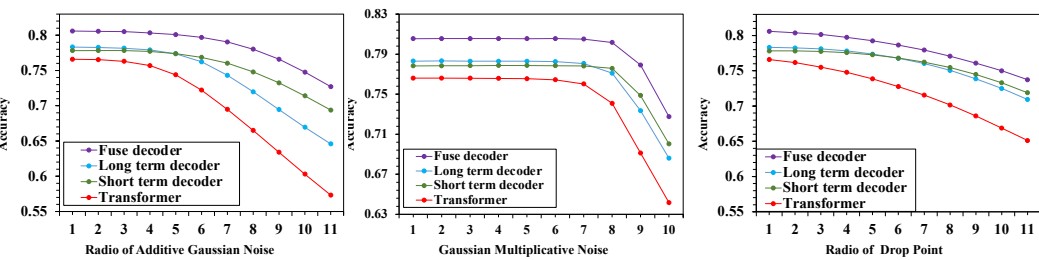

Figure 3: Accuracy versus Noise Intensity Curve.

### 5.1.1 Evaluation Metrics

Following the prior worksGuo et al. (2021); Wang et al. (2020); Chen et al. (2021); Rahimian et al. (2021), we choose the below metrics to evaluate the model's performance. **Pearson Correlation Coefficient** (CC) is a widely used measure of the linear relationship between two variables. It ranges from -1 to 1, where a larger CC value indicates greater similarity between the predicted and estimated joint angles curve, indicating improved estimation. **Root Mean Square Error** (RMSE) is a common metric for evaluating the deviation between predicted and observed values. As the range of fluctuations in the curves of different joint angles can vary significantly, it is difficult to evaluate the performance of models using RMSE alone fairly. Normalization of RMSE addresses this issue, resulting in the Normalized RMSE (NRMSE). **Average curvature** ($\kappa$) of all points for each joint is used to measure the smoothness of an estimated curve. A smaller $\kappa$ indicates a smoother curve.

### 5.2 Comparison with Baselines

We compare the accuracy (ACC) and Standard deviation (STD) between our proposed STET and previous sEMG-based gesture recognition methods. Specifically, we train the model on the GRAB-Myo dataset Pradhan et al. (2022) (the detail of data processing shown in 3.1 ) and separately report the classification results on the categories of Single-finger gestures, Multi-finger gestures, Wrist gestures, Rest, and the overall results. The ratio of training set to test set for each gesture is 5 to 2.

From the experimental results, we can observe that STET consistently performs best on four categories of gestures and overall data. In particular, STET and Zerveas et al. (2021) both use Transformer-based encoders. In the decoder part, STET introduces both the short-term and long-term decoder rather than the fully connected layers used in Zerveas et al. (2021). As a result, the overall accuracy of STET is improved from 80.43% to 90.76% compared to Zerveas et al. (2021). This is because the proposed long-term and short-term decoupling module can extract both the global and fine-grained dependency on the signal dependency and thus can learn better sEMG representations.

Among the transformer-based methods, Informer and STET performed best, with accuracy rates of 87.71% and 90.76%, respectively. Informer relies heavily on max pooling layers to aggregate features, leading to the relative weakness in extracting some short-term features. STET enhances accuracy and stability by strengthening the short-term feature extraction. The improvement is remarkable on the Rest gestures, where the accuracy improves from 83.56% to 95.33%. Furthermore, after incorporating our designed short-term encoder into Informer, its accuracy rate increased from 87.71% to 89.14%, and the classification accuracy for Rest gestures improved from 83.56% to 90.33%.

Note that the Rest category of gesture, as the resting state of devices such as interactive bracelets, is the most frequent gesture that appears in the signals. Thus, the stability of its prediction plays a significant role in the problem. STET achieves the highest and most stable results on the prediction of the Rest category compared with all baselines, indicating the robustness of STET.

### 5.3 Ablation Studies

To validate the effects of the unsupervised sEMG Intrinsic Pattern Capture (EIPC), Long-term decoder, Short-term decoder, Fuse strategy, and loss function. We designed variants of STET and reported their results in Table 3.

First, we can observe that, with the unsupervised EIPC, the accuracy of the transformer and STET is improved by 0.60% and 1.12% compared with training from scratch. This suggests that un-

supervised EIPC can aid in discerning additional data features, such as the inherent variability in sEMG, without the requirement for new samples or extra annotations. Significantly, this process circumvents the need for external data, thus preserving user privacy in the context of data acquisition and processing. Replacing the fully connected layer of the transformer's decoder with the long-term decoder or short-term decoder, the performance is improved by 1.16% and 1.39%, respectively. Furthermore, the performance is comparable when using the long-term decoder or short-term decoder alone, indicating that the two kinds of features may play different significant roles in the sEMG signal recognition, and the short-term cannot be ignored. Most importantly, when we used our designed fuse module to combine long-term and short-term features, the accuracy further improved by 2.46%. This suggests that the two decoders are complementary and that enhancing short-term features is necessary on the basis of the long-term decoder. We employed Asymmetric Loss (ASL) to drive the model's focus on difficult samples. Compared to simply using the Cross-Entropy Loss (CEL), the accuracy improved by 0.73%, indicating the effectiveness of ASL.

| Backbone | In STET framework | AG noise | MG noise | Signal loss |
|---|---|---|---|---|
| Transformer | No | 25% | 16% | 14% |
| Transformer | Yes | 10% | 10% | 8% |
| Informer | No | 11% | 9% | 26% |
| Informer | Yes | 9% | 8% | 17% |

Table 2: Drop rates of accuracy calculated by $\text{drop rate} = \frac{\text{ACC}_{\text{raw}} - \text{ACC}_{\text{noise}}}{\text{ACC}_{\text{raw}}}$. AG: Additive Gaussian noise, MG: Multiplicative Gaussian noise.

## 5.4 ROBUSTNESS ANALYSIS

| Transformer | EIPC | LT | ST | Fusing | CEL | ASL | ACC |
|---|---|---|---|---|---|---|---|
| ✓ | | | | | ✓ | | 85.73% |
| ✓ | ✓ | | | | ✓ | | 86.33% |
| ✓ | ✓ | ✓ | | | ✓ | | 88.02% |
| ✓ | ✓ | | ✓ | | ✓ | | 87.72% |
| ✓ | ✓ | ✓ | ✓ | ✓ | ✓ | | 89.37% |
| ✓ | | ✓ | ✓ | ✓ | | ✓ | 89.42% |
| ✓ | ✓ | ✓ | ✓ | ✓ | | ✓ | 90.54% |

Table 3: The results of ablation study. EIPC: sEMG Intrinsic Pattern Capture; LT: Long-Term decoder; ST: Short-Term decoder; CEL: Cross Entropy Loss; ASL: Asymmetric Loss.

To verify the robustness of the model, we only used high-quality data collected in the lab to train the model and added different types of noise (Additive Gaussian noise, Multiplicative Gaussian noise, and signal loss) during validation to simulate complex scenarios that might be encountered in real situations.

Additive noise typically refers to thermal noise, which is added to the original signal. This type of noise exists regardless of the presence of the original signal and is often considered the background noise of the system in sEMG acquisition. Multiplicative noise is generally caused by channel instability, and it has a multiplicative relationship with the original signal. Also, we simulated signal loss during transmission by randomly setting a portion of the signals to zero.

Figure 3 illustrates the influence exerted by three distinctive noise categories, namely additive noise, multiplicative noise, and signal loss, on the accuracy of the proposed model. The model using only the short-term decoder is less affected by noise compared to the long-term version. This relative robustness of the short-term decoder is potentially attributable to its unique capability to mitigate the global impact of noise by virtue of a sliding window multiple-sampling scheme, which effectively confines the sphere of noise impact. The model that integrates both long-term and short-term characteristics persistently outperforms models that rely on only one. This highlights the significant effectiveness of the integrating process in dealing with noise-induced interference. As depicted in Table 2, it is evident that both Transformer and Informer models demonstrate a notable enhancement in noise resistance when their adopt the design from STET. More details are in the Appendix.

## 5.5 VISUALIZATIONS

To demonstrate the distinction, we first obtained STET's long-term, short-term, and fuse embeddings. The embeddings with dimensions $(N, T, H)$ were then flattened to $(N, T * H)$ and separately projected in 2D by t-SNE, the result shown in Figure 4. We colored each node by category for the illustration. As shown in Figure 4, The classification boundary generated by the long-term feature

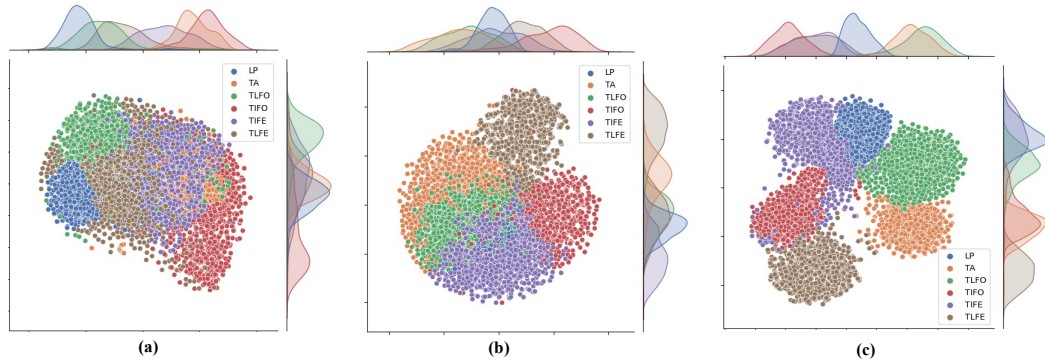

(a)                                  (b)                                  (c)

Figure 4: Visualization of (a) the long-term sEMG embeddings, (b) the short-term sEMG embeddings, and (c) the fused sEMG embeddings for gesture recognition. Note that we color each sample by its classes.

and the short-term feature is a significant difference, indicating that the long-term and short-term features are capable of recognizing different types of gestures. This further suggests that the two features are complementary in data representation. For example, short-term embedding can distinguish TA gesture and TIFO gesture very well, but TA gesture and TIFO gesture will be confused in long-term embedding. Meanwhile, long-term embedding can distinguish LP gesture and TIFE gesture very well, but short-term embedding will confuse them. As shown in Figure 4(c), after the fusion of the two types of features, the classification interface is wider, and the confusion points are significantly reduced, which indicates that the fusion module can effectively complement the strengths of the two types of features.

## 5.6 REGRESSION: HAND JOINT ANGLES PREDICTION

| Model | PCC | NRMSE | $\kappa$ | Time Cost/epoch(s) |
|-------|-----|-------|----------|--------------------|
| LSTM | 0.779 | 0.096 | 0.581 | 26.36 |
| TCN | 0.833 | 0.088 | 1.533 | **3.62** |
| BERT | 0.867 | 0.077 | 1.571 | 4.95 |
| sBERT-OHME | 0.869 | 0.076 | 0.532 | 4.96 |
| STET | **0.877** | **0.073** | **0.522** | 6.83 |

Table 4: Performance comparison of different models. PCC: Pearson Correlation Coefficient, NRMSE: Normalized Root Mean Squared Error.

STET can conveniently handle regression tasks by changing the loss function to mean squared error (MSE) loss. Continuous motion estimation extracts continuous motion information, such as joint angles and torques, from sEMG signals. Since continuous motion estimation requires outputting subtle variations of the movement at each time instant, the local signal variations are particularly important for this type of estimation. In this section, we have re-selected the most competitive models known for sEMG-based joint angle prediction as the baseline and tested the performance of STET on the regression task of predicting the main 10 joint angles for fingers using the Ninapro DB2 Atzori et al. (2014) dataset. As shown in Table 4, STET achieved the best performance in PCC, NRMSE, and $\kappa$, indicating that the joint angle curve predicted by STET is more in line with the real curve and has less abnormal fluctuations, which will significantly improve the user's interactive experience. In terms of training time, due to the addition of the short-term decoder, its training speed is slightly slower than BERT but still within an acceptable range.

## 6 CONCLUSION

Current sEMG-based gesture recognition models usually fail to handle various noisy and distinguish similar gestures, especially in non-laboratory settings. In this paper, we found using short-term information and self-supervised EIPC mitigates this issue. Therefore, we proposed STEM for capturing local signal changes and enhancing noise resistance. The STEM is easily deployable and serves as a plug-in that can potentially be applied to most time series deep learning models. According to our experimental results, our method significantly improved performance for both classification and regression tasks in sEMG, and the model's ability to resist signal loss, Gaussian additive noise, and Gaussian multiplicative noise was clearly improved. This will further drive the practical application of sEMG in VR, AR, and other human-computer interaction scenarios.

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

## A  APPENDIX

### A.1  THE DETAILS OF METHODS

#### A.1.1  THE DETAILS OF sEMG SIGNAL MASKING

Detail of Rationale Behind Our Mask Methods:

***Sensor-wise Mask:*** Each electrode has distinct conditions (e.g., moisture level, coating thickness), resulting in varying signal quality. In real-world scenarios, many types of noise are sensor-specific (e.g., signal loss from certain electrodes or poor electrode contact).

***Contiguous Masked Segments:*** The primary frequency range of sEMG signals is 20–500 Hz, while dataset sampling rates are typically around 2000 Hz. This results in similar values for adjacent sampling points. Contiguous masking prevents the model from relying on adjacent sampling points to predict the masked values.

***The Length of Masked Segments Follows a Geometric Distribution:*** The effective frequency range of sEMG signals varies within 20–500 Hz. Frequency changes manifest as variations in the number of effective sampling points in the time domain. Using dynamic-length masked segments better captures these variations. Geometric distribution aligns with the characteristics of electrode distribution and the frequency dynamics of sEMG signals

The choice of a sensor-wise masking strategy aligns well with the practical scenarios of sEMG noise, where signal disturbances often occur at the level of individual sensors. For instance, signal loss from a specific electrode or cases where electrodes become detached or misaligned frequently affect specific sensor data. By employing sensor-wise masking, our approach effectively mimics these real-world noise conditions. Furthermore, it encompasses scenarios of multi-sensor signal loss, broadening the generalizability of the model across diverse noise conditions. This strategy encourages the model to focus on learning the intrinsic patterns of sEMG signals from unmasked sensors, enhancing its ability to infer missing data. Unlike other masking techniques, sensor-wise masking does not overly depend on spatial-temporal uniformity across sensors, which is crucial given the localized nature of many sEMG disturbances. This robustness ensures better generalization across various signal conditions, as demonstrated by our model's improved performance in handling

---

**Algorithm 1:** The Algorithm of sEMG Signal Masking

---

**Input:** The length of the input signal sequence $t$,
      The number of signal sensors $c$,
      The average length of masked segments $l_m$,
      The masked radio $r$.
**Output:** The mask matrix $\mathbf{M}$.

1 **for** $i = 1, ..., c$ **do**
2    $\mathbf{m}_i = [True] * h$;
3    $p_m = \frac{1}{l_m}$; // probability of each masking segment stopping.
4    $p_u = p_m * r/(1-r)$; // probability of each unmasked segment stopping.
5    $\mathbf{p} = [p_m, p_u]$;
6    $state = Bool(random(0,1) > r)$; // the first state.
7    **for** $j = 1, ..., t$ **do**
8       $\mathbf{m}_{i,j} = state$;
9       **if** $random(0,1) < \mathbf{p}[state]$ **then**
10          $state = \neg state$;

11 **Return** $\mathbf{M} = ([\mathbf{m}_i]_{i=0}^c)^T$;

---

noisy and partial signals. By addressing practical noise issues systematically, the proposed approach enhances the model's resilience in real-world applications.

### A.1.2 ROLE OF UNSUPERVISED PRETRAINING AND NOISE ROBUSTNESS

our sEMG Signal Masking strategy aims to capture intrinsic variability in the signal rather than explicitly removing noise. This improves the model's robustness by:

1) Preventing overfitting on spurious patterns in supervised training.

2) Enabling better feature learning from unlabeled data.

As shown in Table 2, this approach reduced performance degradation under various noise conditions (e.g., additive Gaussian noise impact decreased from 25% to 10%).To further understand the contribution of the pretrain with sEMG Signal Masking strategy, we conducted additional experiments to evaluate its impact on the model's robustness to various noise types. Specifically, we removed the short-term and long-term enhancement modules and assessed the backbone model (Transformer) under conditions with and without pretraining using the sEMG Signal Masking strategy. Other experimental conditions, such as ASL loss, remain unchanged. Due to time constraints, this experiment was conducted on data from only three subjects. The results are summarized below:

### A.1.3 THE DETAILS OF EXPERIMENT ON TABLE 2

In this experiment, we utilized transformers and informers as the backbone network, employing the same parameters as in Table 1. The STET framework incorporates a pretraining strategy with sEMG masks and STEM on top of the backbone. In contrast, the No STET framework uses a standard pretraining with a general mask, employing a standard masking approach with fixed segment lengths of 3 (aligning with the average length in our method), sensor-agnostic masking, and a random masking ratio of 15% (consistent with our method). Other experimental conditions, such as ASL loss, remain unchanged.

| Backbone | Pretrain with sEMG Signal Masking Strategy | AG Noise | MG Noise | Signal Loss |
|---|---|---|---|---|
| Transformer | No | 22% | 15% | 16% |
| Transformer | Yes | 15% | 13% | 12% |

Table 5: Comparison of performance with and without sEMG Signal Masking Strategy.

### A.1.4 The definition of long term encoder

$$\text{MultiHead}_L\left(\mathbf{X}^{(L)}\right) = \text{Concat}\left(h_1, \ldots, h_d\right)\mathbf{W}^O, \tag{4a}$$

$$\text{where} \qquad \{h_i\}_{i=0}^d = \{\text{Attention}\left(\mathbf{Q}_i, \mathbf{K}_i, \mathbf{V}_i\right)\}_{i=0}^d, \tag{4b}$$

$$\text{Attention}\left(\mathbf{Q}_i, \mathbf{K}_i, \mathbf{V}_i\right) = \text{Softmax}\left(\frac{\mathbf{Q}_i\mathbf{K}_i^T}{\sqrt{h}}\right)\mathbf{V}_i, \tag{4c}$$

where $\mathbf{Q}_i = \mathbf{X}^{(L)}\mathbf{W}_i^Q, \mathbf{K}_i = \mathbf{X}^{(L)}\mathbf{W}_i^K, \mathbf{V}_i = \mathbf{X}^{(L)}\mathbf{W}_i^V$ and $\{\mathbf{W}_i^Q, \mathbf{W}_i^K, \mathbf{W}_i^V\}_{i=0}^d \in \mathbb{R}^{h \times h}$ are parameter matrices and $d$ is the number of attention heads. $\text{Concat}(\cdot)$ represents the concatenate operation. $\mathbf{W}^O \in \mathbb{R}^{dh \times h}$ is the output parameter matrix to transform the concatenated outputs of $d$ attention heads. Then, the long-term sEMG embeddings $\mathbf{H}^l \in \mathbb{R}^{t \times h}$ is obtained by $\mathbf{H}^l = \text{MultiHead}_L(\mathbf{X}^{(L)})$.

The linear projection is applied in the long-term enhanced module because this module processes inputs of fixed length due to its structural design. We use linear projection here to adjust the feature dimensions appropriately. In contrast, the short-term enhanced module operates based on a sliding window, which means the input length remains consistent and does not require dimensional adjustment. Therefore, linear projection is not needed in the short-term module.

### A.2 The significance of the results

We conducted Friedman and Wilcoxon signed-rank tests to analyze significant differences among subjects with different evaluation methods, and we corrected the P-value using Bonferroni correction. Our model still outperformed the other models listed in Table 1 ($P < 0.01$). Finally, we would like to emphasize that in sEMG-related HCI applications, even a small improvement in accuracy can lead to a significant improvement in user experience. Therefore, we believe that our proposed model can offer a significant improvement in user experience.

### A.2.1 The details of noise

Additive noise typically refers to thermal noise, which is added to the original signal. This type of noise exists regardless of the presence of the original signal and is often considered the background noise of the system in sEMG acquisition. Additive noise can be described as:

$$G_{add}(x) = x + \alpha \cdot N(x), \tag{5}$$

$$N(x) = \frac{1}{\sqrt{2\pi}}exp(-\frac{(x-u)^2}{2\sigma^2}), \tag{6}$$

where $G(x)_{add}$ represent the signal with additive noise, $\alpha$ is used to adjust the size of the noise. $N(x)$ is a normal distribution that simulates background noise. Here, we set $u$ to 0 and $\sigma$ to 1.

Multiplicative noise is generally caused by channel instability and has a multiplicative relationship with the original signal. Multiplicative noise used in the experiment can be described as:

$$G_{mul}(x) = x + N_{mul}(x), \tag{7}$$

$$N_{mul}(x) = n \cdot \frac{x^2}{N(x)^2} \setminus 10^{\frac{SNR}{10}}, \tag{8}$$

where $G(x)_{add}$ represent the signal with multiplicative noise, $SNR$ stand for the signal to noise ratio.

Additionally, we simulated signal loss during transmission by randomly setting a portion of the signals to zero.

### A.3 About the robustness of noise

Our short-term enhancement can be easily extended to other models. It is evident that both Transformer and Informer models demonstrate a notable enhancement in noise resistance when their decoders are replaced with the design from STET. Specifically, when comparing the drop rates of accuracy due to different noise types, calculated by $rate = \frac{ACC_{\text{raw}} - ACC_{\text{noise}}}{ACC_{\text{raw}}}$:

- **Transformer**:

  Without the STET framework, it experienced a drop of 25% under additive Gaussian noise, 16% under multiplicative Gaussian noise, and 14% with signal loss. However, when integrated into the STET framework, these drops were reduced to 10% for both additive and multiplicative Gaussian noise and 8% for signal loss.

- **Informer**:

  Without the STET framework, it showed a drop of 11% due to additive Gaussian noise, 9% due to multiplicative Gaussian noise, and a significant 26% with signal loss. With the STET design, these rates improved to 9% for additive Gaussian noise, 8% for multiplicative Gaussian noise, and 17% for signal loss.

These results highlight the efficacy of the STET framework in enhancing the robustness of both Transformer and Informer models against various noise types.

### A.3.1 REAL-WORLD DEPLOYMENT AND HUMAN-SUBJECT STUDY

Although most previous works, including ours, have primarily tested the performance of gesture detection algorithms on offline laboratory datasets Wang et al. (2020); Chen et al. (2021), we recognize the importance of real-world application scenarios. These scenarios often involve complex variables such as electrode movement and muscle state changes, which are not adequately captured in offline testing. To address this gap, as shown in Figure 1, we expanded our evaluation to include online performance verification. This was achieved by integrating our algorithm with a 3D virtual hand, developed using the Unreal 5 engine, and controlling it through STET decoding of sEMG signals.

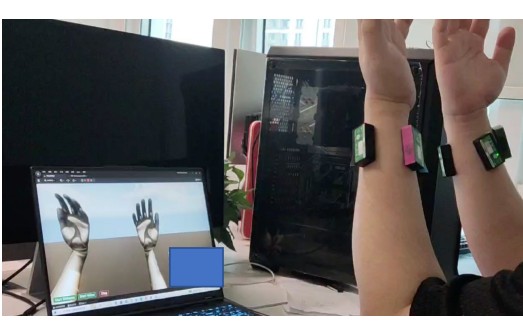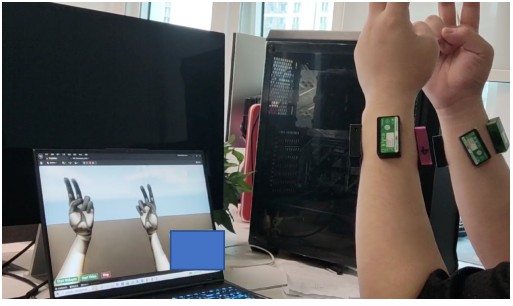
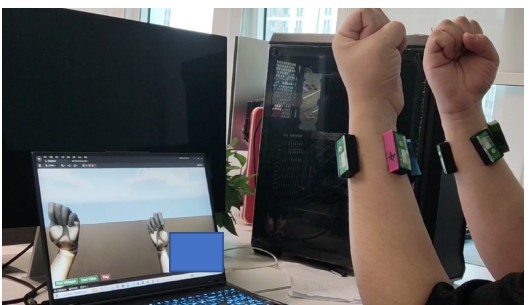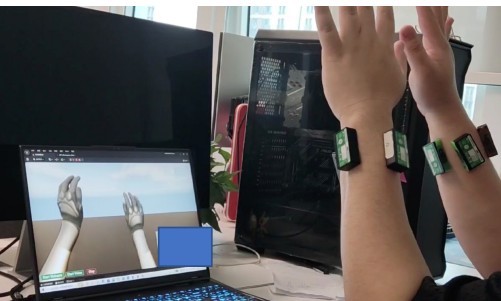

Figure 5: STET-based Real-Time Hand Interaction Reconstruction from Wrist sEMG Signals.

To make our system more aligned with daily usage habits, we placed four myoelectric electrodes on both the left and right wrists of users, each with a sampling rate of 2000Hz. The signals were transmitted to the host computer via a WiFi interface for continuous motion estimation. Our online experiment results demonstrated an overall latency of less than 50ms and a Pearson Correlation Coefficient (PCC) of over 0.8. This indicates a stable, accurate, and natural interactive experience, surpassing the capabilities of computer vision-based methods in terms of energy efficiency and independence from lighting conditions and occlusions.

To enable streaming processing online, we implemented an online buffer mechanism. Specifically, the buffer collects incoming signals in real-time, and every 10ms, the model retrieves the most recent 200ms of signals from the buffer for processing. The architecture used for online testing is identical

to the one described in the paper. Furthermore, the model has been deployed in a commercial sEMG interaction application, where it has undergone testing in VR control scenarios, demonstrating its practical usability.

In our human-subject study, we recruited a total of eight healthy participants, comprising an equal gender distribution of four males and four females, all of whom were right-handed. Each participant was asked to perform six distinct movements, with these movements being captured using both a standard transformer model and our STET model. Importantly, the participants were blinded to the model in use during their tasks. After engaging in a 10-minute gaming session designed to test the models, participants were asked to identify which model they felt was more stable and provided a better experience. Remarkably, out of the eight participants, seven preferred the STET model, citing its greater stability and overall performance. This overwhelming preference for the STET model amongst participants highlights its efficacy and potential for real-world applications, reinforcing our findings regarding its superior performance compared to traditional models.

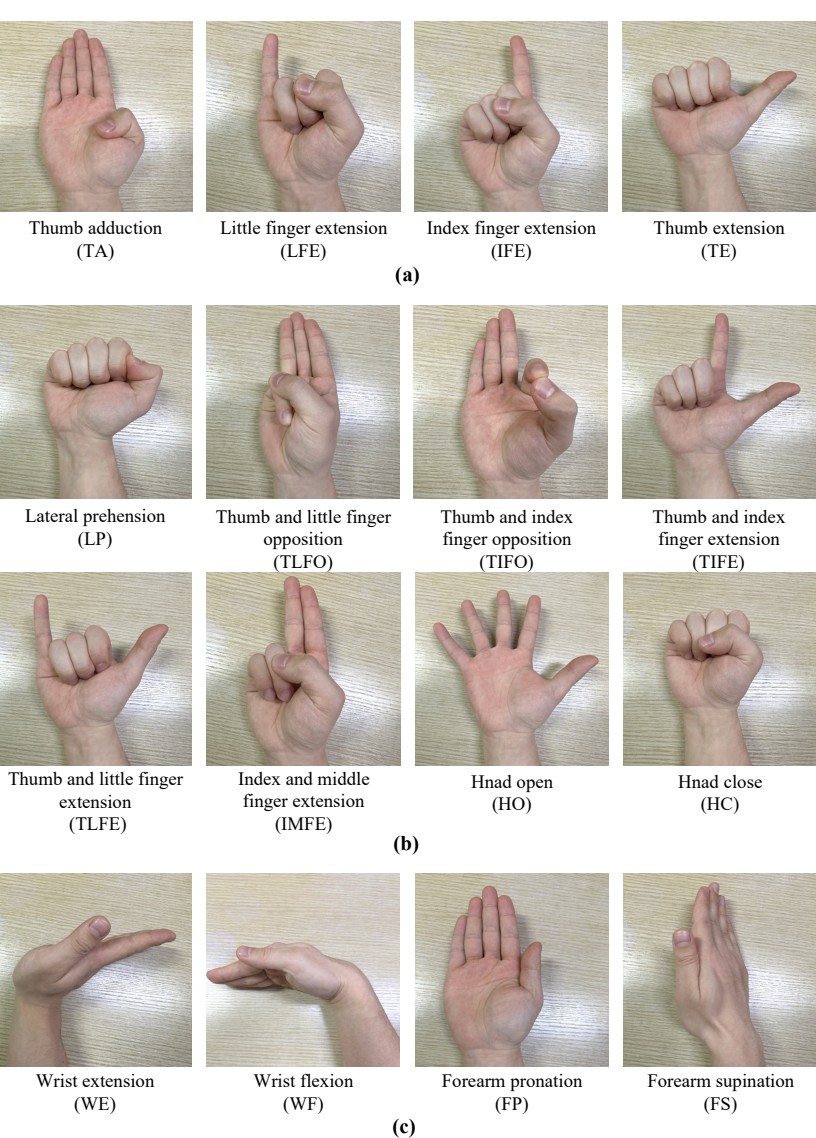

Figure 6: The gestures were used in gesture classification experiment:(a) Single-finger gestures (b) Multi-finger gestures (c) Wrist gestures.

## B    THE DIFFERENCE FROM LST-EMG-NET

Our proposed approach exhibits a marked deviation from the LST-EMG-Net as detailed in Zhang et al. (2023). In the LST-EMG-Net, the raw sEMG data is pre-segmented into long and short durations prior to the network ingestion. This pre-segmentation could potentially hinder the encoder's capability to grasp the intrinsic patterns of sEMG. In contrast, our method divides the data into short-term and long-term segments during the decoding process, which we believe is a more effective strategy. Our results support this, with our method achieving an accuracy of 90.8% compared to 85.3% achieved by the LST-EMG-Net model.

## C    DATASET

We conducted the experiments on the Gesture Recognition and Biometrics ElectroMyogram (GRABMyo) Dataset, the largest known open-source wrist EMG dataset with great potential for developing new generation human-machine interaction based on sEMG. GRABMyo has 43 healthy subjects whose average age is $26.35 \pm 2.89$, and the average forearm length is $25.15 \pm 1.74$ cm (measured as the distance between the olecranon process and the ulnar styloid process). There were 23 male subjects and 20 female subjects, respectively.

*Data collection.* Place 2 rings of 12 monopolar sEMG electrodes (AM-N00S/E, Ambu, Denmark) at the wrist position; each ring consists of 6 electrodes. The sample rate was set to 2048Hz. The distance between the center lines of adjacent electrodes is 2cm. In order to keep the electrode positions consistent across subjects, the position of the first electrode was fixed at the centerline of the elbow crease. The subjects performed 17 gestures of hand and wrist (including a rest period sEMG) according to the prompts on the computer screen. Each gesture was repeated seven times, each lasting 5 seconds. To avoid muscle fatigue, rest 10s between repetitions. In the following experiments, we use five repetitions as the training set and two repetitions as the test set.

*Data processing.* To avoid muscle fatigue, rest 10s between repetitions. In the following experiments, we use five repetitions as the training set and two repetitions as the test set. Bandpass filtered between 10 Hz and 500 Hz with a gain of 500 was adopted to the raw signal. We use the difference between the corresponding electrodes in the two loops as the input signal of our model.

*Sizes of Training, Validation, and Test Sets.* The TableC shows the division of the training set, validation set, and test set, which aligns with the common comparison practices in the field. Note that after selecting the parameters using the validation set, the validation set will be merged into the training set for retraining. **Gesture Classification Task:** For this task, we used the \*\*GRABMyo\*\* dataset for both pre-training and fine-tuning. The GRABMyo dataset provides the necessary data for gesture classification, but it does not include joint angle information. **Hand Joint Regression Task:** Since the GRABMyo dataset lacks joint angle data, we employed the Ninapro DB2 dataset for this task. Both pre-training and fine-tuning were conducted using data exclusively from Ninapro DB2.

| Task Type | Data Used | Samples of each user (except rest) |
|---|---|---|
| **Classification** | | |
| Training Set | Five repetitions per gesture per user | ~42,500 samples |
| Validation Set | One repetition randomly selected from the training set | ~8,500 samples |
| Test Set | Two repetitions per gesture per user | ~17,000 samples |
| **Regression** | | |
| Training Set | Four repetitions per gesture per user | ~34,000 samples |
| Validation Set | One repetition randomly selected from the training set | ~8,500 samples |
| Test Set | Two repetitions per gesture per user | ~17,000 samples |

Table 6: Data Splits for Classification (GRABMyo) and Regression (Ninapro DB2) Tasks

*Participant-Dependent Experiments.* Our experiments are participant-dependent (user-specific). For each user, we train and evaluate the model using data exclusively from that user. This approach aligns with the objectives of our study, focusing on enhancing the model's robustness and performance for individual users without introducing additional data from other participants during pre-training or fine-tuning. This is a common practice in sEMG studies when aiming to optimize performance for specific users.

## C.1 SUPPLEMENTARY DETAILS OF THE EXPERIMENTAL SECTION

All comparison methods reported in our paper were re-implemented by us. This decision was made because different methods in the literature often use different datasets, select varying subsets of gesture categories, or employ different evaluation protocols, making direct comparisons challenging. By re-implementing these methods under a consistent framework, we ensured that all methods were evaluated:

1) On the same dataset(s)

2) Using the same set of gesture categories

3) Under identical training, validation, and testing conditions

4) With consistent hyperparameter tuning strategies based on the original papers

### C.1.1 FAIRNESS OF COMPARISON REGARDING PRETRAINING

We want to emphasize that the pretraining in our work is fundamentally different from typical pre-training approaches in fields like NLP. Unlike models that are pretrained on large external datasets and then fine-tuned on specific tasks, our method performs pretraining without introducing any external data. We use only the user-specific training data for both pretraining and fine-tuning. This means that both our method and the comparison methods are trained and tested on exactly the same data, ensuring a fair comparison. Our contribution lies in demonstrating that even without additional data, our approach enhances the model's robustness and performance using the existing user-specific data.

### C.1.2 EVALUATION PROTOCOL COMPARED TO THE ORIGINAL DATASET PAPER

The original GRABMyo dataset paper did not directly evaluate gesture classification accuracy or provide corresponding benchmarks for classification tasks. Instead, the dataset's quality was assessed using metrics like the Area Under the Curve (AUC) and Equal Error Rate (EER). Therefore, our work does not follow the exact evaluation protocol proposed in the original paper because such a protocol for gesture classification was not established.

### C.1.3 DETAILS OF EVALUATION METRICS

**Root Mean Square Error** (RMSE) is a common metric for evaluating the deviation between predicted and observed values. As the range of fluctuations in the curves of different joint angles can vary significantly, it is difficult to evaluate the performance of models using RMSE alone fairly. Normalization of RMSE addresses this issue, resulting in the Normalized RMSE (NRMSE).

$$\text{RMSE} = \sqrt{\sum_{i}^{N} \frac{(\theta_{\text{est}} - \theta_{\text{real}})^2}{N}},$$

$$\text{NRMSE} = \frac{\text{RMSE}}{\theta_{\max} - \theta_{\min}}$$

where $\theta_{\text{est}}$ and $\overline{\theta_{\text{est}}}$ are the estimated angle and their average, while $\theta_{\text{real}}$ and $\overline{\theta_{\text{real}}}$ are the real angle and their average. $\theta_{\max}$ is the maximum of the real angle, and $\theta_{\min}$ is the minimum of the real angle.

**Pearson Correlation Coefficient** (CC) is a widely used measure of the linear relationship between two variables. It ranges from -1 to 1, where a larger CC value indicates greater similarity between the predicted and estimated joint angles curve, indicating improved estimation.

$$\text{CC} = \frac{\sum_{i=1}^{N} \left(\theta_{\text{est}} - \overline{\theta_{\text{est}}}\right) \left(\theta_{\text{real}} - \overline{\theta_{\text{real}}}\right)}{\sqrt{\sum_{i=1}^{N} \left(\theta_{\text{est}} - \overline{\theta_{\text{est}}}\right)^2} \sqrt{\sum_{i=1}^{N} \left(\theta_{\text{real}} - \overline{\theta_{\text{real}}}\right)^2}} \tag{9}$$

## C.2 Inference Performance and Parameter Comparison of Models

| Model | Inference Time-GPU (A6000) | Inference Time-CPU (AMD EPYC 7543) | Parameter Count | GPU Memory Allocated |
|---|---|---|---|---|
| Transformer | 3.8 ms | 15.1 ms | 481169 | 18.08 MB |
| Add STEM with weight sharing | 3.9 ms | 17.6 ms | 489233 | 23.66 MB |
| Without weight sharing | 4.8 ms | 27.5 ms | 581137 | 21.65 MB |

Table 7: The comparison of inference time, number of parameters, and GPU usage between the model using STEM and a non-weight sharing transformer layer

In Table 7, we use following high paramter: Feature dimension is 12, maximum length is 200, model dimension is 64, number of attention heads is 2, number of layers is 3, dimension of feedforward network is 256, number of classes is 17, dropout rate is 0.1, positional encoding is 'learnable', activation function is 'gelu', and normalization is 'BatchNorm'. The STEM model uses the same parameters as those in the experimental setup of the paper.

**Parameter Count:** By incorporating STEM, we significantly reduce the model's parameter count. Specifically, when the STEM module is added (with weight sharing enabled), the parameter count increases slightly from 481,169 to 489,233, an increase of about 1.7%. However, if weight-sharing is not used, the parameter count increases substantially to 581,137, a 21% increase.

**Inference Time:** We measured the inference time on both GPU and CPU. When using the STEM module with weight sharing, the GPU inference time is 3.9ms, and the CPU inference time is 17.6ms, which is a significant improvement compared to the case without the STEM module (4.8ms on GPU and 27.5ms on CPU). This indicates that the STEM module, while enhancing short-term features, maintains a low inference time.

**GPU Memory Consumption:** The relative increase in GPU usage is due to the mechanism of parallel computation that the GPU activates when using a sliding window.

### C.2.1 Parameters search and the model's sensitivity

The selection of the masking ratio (0.15) and the average length of masked segments was informed by a systematic hyperparameter search using the (0.05, 0.15, 0.25, 0.35, 0.45), (1,3,5,7,9) range. This was conducted via wandb, ensuring a comprehensive evaluation of the model's sensitivity to these parameters. We observed that a masking ratio of 0.15 and average length of 3 consistently yielded the best performance in terms of classification accuracy. While the accuracy fluctuated by approximately 4% across the tested range, the 0.15 ratio provided an optimal balance between introducing sufficient noise for robust feature learning and retaining enough original signal for effective pretraining. The choice of segment length further complements this masking strategy, as it ensures that the model learns to reconstruct meaningful patterns while not overly relying on adjacent unmasked data. These parameters jointly enable the model to focus on capturing intrinsic signal variability, thereby enhancing its resilience to real-world noise scenarios. Such robustness aligns well with our goal of improving generalization in noisy environments.

Regarding window sizes of STEM, we experimented with varying configurations and found that a short-term window size of 41, 21 with a step size of 1 provided the best balance between noise isolation and feature granularity. We use wandb to search the best window size, the search space is (11,21,31,41,51,61). the accuracy fluctuated by approximately 3% across the tested range. In the future, we will collaborate with biologists to explore the relationship between window size and different gesture classifications on a biological level.

