# OpenReview forum: "Revisiting Noise Resilience Strategies in Gesture Recognition: Short-Term Enhancement in Surface Electromyographic Signal Analysis"
_ICLR.cc/2025/Conference — Submitted to ICLR 2025_

### Official Review · Reviewer_aqm7 · 2024-10-31

**Soundness:** 3
**Presentation:** 3
**Contribution:** 3
**Rating:** 6
**Confidence:** 4

**Summary:**

This paper addresses the task of gesture classification and hand joint angle prediction from sEMG signal from a multi-electrode array. Specifically, the authors make three main contributions: 1)  Self-supervised pre-training based off of a signal masking objective 2) The introduction of short-term and long-term modules which are fused to create a final prediction 3) Use of asymmetric loss function to address asymmetries to address the imbalance between positive and negative samples.
The self-supervised pre-training step enables training on unlabeled EMG signals which is valuable given the sparse state of labeled EMG datasets. The self-supervised objective consists of reconstruction of masked signals, its loss denoted by MSE. The masks were constructed such that contiguous chunks of signal are masked at once, as otherwise neighboring signal timestamps make the objective too simple.
The short-term and long-term modules are both transformer-based models. The main difference between the two is that for the short-term module, the attention is limited within a certain window size. This forces the model to focus only on this short-term context to perform predictions.
The asymmetric loss follows Ridnik et al., and was introduce after making the observation that the positive and negative samples given a gesture is severely unbalanced in favor of the negatives.
Experimental results demonstrate improved performance over related work. Ablation studies show improved robustness to artificially added noise, as well as the improvement each proposed component gives.

**Strengths:**

- Thorough empirical investigation of their proposed approach with various ablation studies
- Comparisons to related work show SotA performance
- Approach tested on both hand joint angle prediction as well as gesture classification
- Statistical significance test performed, which isn't common but valuable (appendix)
- Combination of their proposed approach with prior works architecture demonstrates generalizability of the improvements (e.g TEMGNet 78% vs TEMGNET+STEM 84%)

**Weaknesses:**

- Although the long term module has been empirically validated to show improved performance I would've liked to see better explanation / investigation on what kind of information is being stored in long term vs short term module. EMG inherently encodes force and gestures are performed on a short time scale. Therefore it is not immediately clear to me why the long term module improves performance
- In similar manner, the ablation study demonstrates the performance of one module over the other and over their combined performance, where the combined performance yields the best results. However, one could also conclude from these results that having an extra module is what lead to the improved performance (i.e extra model parameters), and not the the type of module. To that end, how would the model perform with two LT or ST units and how does that compare to the fused variant?

**Questions:**

- L035-036: motoring -> monitoring
- L240-241: slide -> sliding
- What kind of information does LT vs ST store?
- Could the author clarify what kind of data pre-training was done on vs. fine-tuning?

---

### Official Review · Reviewer_dCsH · 2024-11-02

**Soundness:** 2
**Presentation:** 2
**Contribution:** 2
**Rating:** 6
**Confidence:** 4

**Summary:**

This paper proposes an innovative solution for improving sEMG-based gesture recognition, tackling issues such as noise interference and distinguishing similar gestures, especially in non-laboratory settings. The authors introduce the Short-Term Enhancement Module (STEM), which captures local signal variations to maintain noise resistance, and a self-supervised sEMG Intrinsic Pattern Capture (EIPC) for pre-training and learning intrinsic signal patterns. STEM is designed to be easily integrated into various time-series deep learning models. Experiments show it significantly improves performance in classification and regression tasks, making it suitable for practical applications.

**Strengths:**

The paper is well-written and structured clearly, making it easy to follow.
The parallel use of both long-term and short-term decoders is an innovative approach that preserves context without losing important information from different stages in the sequence. This approach helps address limitations in existing sEMG models and presents a new way of combining local and global signal features, adding originality and significance to the field.
The self-supervised pre-training method effectively reduces the need for extensive labeled data and boosts the model's generalization, making it practical for real-world applications.
Additionally, the emphasis on noise robustness, a common challenge in sEMG signal processing, enhances the model's real-world applicability.

**Weaknesses:**

Insufficient Methodological Rationale: The paper would benefit from more detailed explanations for certain design choices, such as the selection of the window size in the short-term decoder and the masking ratio in the EIPC module. Clarifying how these parameters were optimized or chosen could strengthen the methodological foundation.

**Questions:**

1. On line 80, the authors state, *“we show that the long-term and short-term features are complementary in sEMG-based gesture recognition tasks.”* While the methodology section explains these concepts, it would improve reader clarity if a brief explanation of short-term and long-term features were included earlier in the paper.
2. Please clarify the rationale behind the choice of a sensor-wise masking strategy over other potential approaches, and how this decision impacts the model’s ability to generalize to various sEMG signal conditions.
3. Please provide more detail on why the specific masking ratio (0.15) and the average length of masked segments were chosen. Also, please add explanation of the sensitivity of the model’s performance to these parameters.
4. Could the authors explain why a multi-head self-attention layer was chosen for the long-term decoder and a sliding-window self-attention layer for the short-term decoder.
5. For the sliding-window self-attention, how was the window size determined, and how sensitive is the model’s performance to variations in this parameter?
6. Please clarify how normalization was performed for the NRMSE (Normalized Root Mean Squared Error) metric.

---

### Official Review · Reviewer_bTGn · 2024-11-03

**Soundness:** 2
**Presentation:** 2
**Contribution:** 2
**Rating:** 5
**Confidence:** 3

**Summary:**

This manuscript proposes a new framework for gesture recognition and regression from sEMG signals. They two existing datasets, GRABMyo and Ninapro that record sEMG alongside gesture identity and finger joint angles and develop a framework, STET, to process the signals. STET combines a pre-trained masked autoencoder with two transformers that are designed to process short term and long term information in parallel. These signals are concatenated, and in the case of gesture recognition, use a new asymmetric loss to upweight positive samples. The approach is thus a combination of novel components, although there is considerable past precedent for these approaches in the literature (eg LST-EMG-Net for separately modeling long and short term context and CutMix for time and channel wise masking of temporal signals). The experiments compare accuracy for gesture classification with a variety of baselines and perform ablations of different network components. They perform separate experiments simulating the addition of noise.

**Strengths:**

- The approach appears to be novel. While I am somewhat surprised that this approach would be better than a single transformer with a variety of temporal scales, it is reminiscent of other work in CV integrating different spatial context.
- The ablations are thorough, and there are numerous baselines.
- The pretraining appears to reduce the effects of additive and multiplicative noise.
- the application domain is timely and interesting

**Weaknesses:**

Cons
- There are a few novel components here, but I don't find the combination especially well developed or compelling. It is likely the case that their combination of masking, parallel coarse and fine transformers and asymmetric losses improves performance but the intuition advances is a little ad hoc. Unsupervised pretraining might help, but likely not by reducing gaussian noise in the signal. It is unclear whether the short of long term components are more important for affecting different types of gestures, it is not clear what window sizes are important. There are experiments oriented at some of these, but they don't quite get to the bottom of things.

- It is unclear if the developed approach can be run online, and how it performs in the most important generalization domain for sEMG, across novel participants. So in practice I am not sure how useful this approach is.

- Given the lack of documentation of hyperparameters for all models (see below) or statistics it is hard to evaluate the results critically.

- The novelty of the long and short timescale approach is questionable. As they note, other manuscripts in the literature use this approach eg L262 claims that these long and short timescales are run in serial in LST-EMG-Net, but based on Figure 3 of Zhang et al 2023 they appear to be run in parallel.

- The added gaussian and multiplicative noise experiments are not especially convincing to me as a realistic noise aggressor for sEMG, where the predominant failure mode is generalization across participants or issues like contact loss or power line interference. Given that the masking approach only improves accuracy by 0.6-1% in the GrabMYO data, it seems like these noise sources are not large. I find the noise discussion is a bit of a digression from the rest of the manuscript.

- The related work section can be better organized, and the novelty and distinction between the architectures can be better presented in the related work, rather than throughout the text and appendix.

**Questions:**

- The data collection done for the robustness analysis is unclear. Is it the experiment described in Figure 6? Similarly, it is unclear if the methods in ‘dataset’ section describe data collected for this work or different work
- How were hyperparameters determined e.g. the short window length, for this dataset? what about the hyperparameters for the other baselines? what was the train/test/validation strategy? Was the data tested on held out users or held out gestures within a user? What fraction of data was used for the unsupervised pretraining?
- How was the model run online in a streaming fashion? Is the architecture benchmarked offline the same as run online?
- Is it the short term or long term e.g. for table 3 what is the breakdown by gesture.

---

> ### Comment · Reviewer_5Rea · 2024-11-24
> **Re**
>
> - The big claims on contribution of the work in Abstract section are major concerns (e.g., inappropriate claim of 'Learnable denoise').
>
>
> - Here is my orginal comment on **inconsistencies** between code and manuscript :  'We pre-train on GRABMyo for 20 epochs using a fixed learning rate of 1e-4 for the backbone.' (line 307-308). The manuscript states that GRABMyo was used for pretraining. However, the code indicates that pretraining was conducted on Ninapro DB2. Please see evidence: (1) https://anonymous.4open.science/r/short_term_semg/pretrain.py, code line 21: train_loader, val_loader = get_dataloader_db2(dataCfg, "EMG_CSV/S1_E1_A1/"); (2) https://anonymous.4open.science/r/short_term_semg/cfg/db2.yaml; (3) https://anonymous.4open.science/r/short_term_semg/dataloaders/Ninapro.py, code line 67, function get_dataloader_db2(cfg, path_s,exercise), which returns data from Ninapro. The descriptions in the manuscript and code appear contradictory and require clarification. Furthremore, the manuscript does not discuss which dataset was used for fine-tuning. Based on the code in https://anonymous.4open.science/r/short_term_semg/finetuning.py, fine-tuning seems to employ two datasets, namely "hospital" and Ninapro DB2.
>
> - **If the authors acknowledge that the code is incomplete, then it must be explicitly stated in the manuscript when submitted.** It is **unacceptable** to submit code for review purposes and subsequently attribute contradictory findings to its incompleteness **after reviewers identifed discrepancies**.
>
> - As a reviewer, I evaluate submissions thoroughly and fairly, focusing on the scientific rigor and clarity of the work presented. While I may not have directly worked on **every** specific aspect of the paper, I evaluate the work on established research principles on representation learning, knowledge in the field of sEMG, as well as **the clarity of the claims**, and the evidence provided in the **code and manuscript.**
>
> - Given the current state of the manuscript, where many major concerns remain unresolved, I **do not recommend acceptance** for ICLR at this time. I urge the authors to thoroughly address all raised concerns of **original and further comments since rebuttal phase** to meet the standards expected for acceptance.

---

### Official Review · Reviewer_Xpzg · 2024-11-04

**Soundness:** 3
**Presentation:** 3
**Contribution:** 3
**Rating:** 8
**Confidence:** 3

**Summary:**

Surface Electromyography (sEMG) is highly susceptible to noise in real-world environments. To enhance noise resilience in sEMG-based models, this paper presents three key contributions:
1) introduction of the Short-Term Enhancement Module (STEM): A scalable, learnable, and low-cost module that enhances noise resistance. When integrated into neural networks, STEM improves performance by focusing on short-term signal features critical for distinguishing gestures amidst noise.
2) Self-Supervised Signal Masking: This technique leverages intrinsic variability in sEMG signals to enhance pre-training, allowing the model to learn robust representations without requiring extensive labeled data.
3) comprehensive Evaluation on the GRABMyo Dataset: Experiments on the largest available wrist sEMG dataset, GRABMyo, demonstrate that the Short-Term Enhanced Transformer (STET) achieves superior accuracy and robustness compared to existing methods, with reduced accuracy drop rates under noise and more precise gesture classification boundaries.

**Strengths:**

1) to address the risk of self-supervised learning fitting to ineffective patterns and missing meaningful temporal semantics, the authors introduce a sophisticated masking strategy. This approach uses a geometric distribution to control masked and unmasked segments, promoting more robust temporal feature learning and improving the model's ability to capture essential sEMG patterns.
2) the authors propose a dual-decoder approach that extracts both long-term and short-term dependencies within the signal sequences. The long-term decoder captures the global context and overall signal structure, while the short-term decoder focuses on specific local characteristics critical for accurate gesture differentiation, enhancing both precision and robustness.
3) an asymmetric loss function is introduced to address class imbalance and improve model optimization by emphasizing difficult samples. This loss function enhances the model's ability to generalize across varied gesture categories, prioritizing accurate classification for challenging cases without overfitting on easy negatives.
4) the paper is well-structured and logically organized, with clear presentation of theorems, statements, and methodological components. The writing is precise and effectively communicates the technical contributions, making the model and findings accessible to readers.

**Weaknesses:**

1) The experimental evaluation presents classification results solely on the GRABMyo dataset, which may introduce dataset bias. To improve the robustness and generalizability of the findings, it would be beneficial to include experiments on additional publicly available datasets.

**Questions:**

1) The authors claim that the proposed STEM module is low-cost; however, the paper lacks a detailed description or analysis of the module's computational efficiency. Could the authors clarify how they evaluated STEM's efficiency and provide comparative metrics or benchmarks to substantiate its low-cost nature? Additionally, how does STEM’s computational cost compare to other common noise resilience approaches?

---

### Official Review · Reviewer_5Rea · 2024-11-05

**Soundness:** 2
**Presentation:** 2
**Contribution:** 1
**Rating:** 3
**Confidence:** 5

**Summary:**

The authors propose a Short Term Enhancement Module (STEM) to improve noise resilience in sEMG-based gesture recognition. The results show that the proposed method performs the best, both with and without additive noise, on the publicly available EMG dataset, GRABMyo.

**Strengths:**

Gesture recognition using sEMG is a significant field with applications in prosthetic control and rehabilitation.  Extensive experiments have been conducted in this work. The inclusion of code is appreciated. However, clearer instructions are needed.

**Weaknesses:**

Major concerns on the Claims and Contributions:
1. 'Learnable denoise, enabling noise reduction without manual data augmentation' (line 022-023).
The authors claim to introduce a "learnable denoise" method, enabling noise reduction without manual data augmentation. To the best of my knowledge, 'learnable denoise' typically refers to a model trained on paired noisy and clean data in an unsupervised manner, as illustrated in Fig. 2 of Ref [1] (published in CVPR'2024). Another common approach of denoising is to employ an autoencoder-based structure. However, this manuscript does not incorporate such a strategy in the pretraining stage. In this work, pretraining stage appears limited to signal or segment masking, as suggested by Equation 1, Figure 2(a), and Algorithm 1 in the Appendix. This is also evident in the provided code: https://anonymous.4open.science/r/short_term_semg/model/generate_mask.py.

2. 'Scalability, adaptable to various models' (line 023).
The claim of "scalability" as described here appears to lack justification. Please correct me if I am wrong, 'scalability' typically refers to a model’s ability to handle increased data or computational demands, which is not evidently addressed here. "Adaptability" may be a more appropriate term in this context.

3. 'Cost-effectiveness, achieving short-term enhancement through minimal weightsharing in an efficient attention mechanism.'  (line 024).
The term "cost-effectiveness" seems ambiguous, with no evidence for computational efficiency. While the authors claims the reduced training time, other metrics such as parameter count, inference time, FLOPs, and GPU memory consumption, are not discussed.

4. Limited novely, 'Intrinsic Variation Capture Module' repeated 14 times in this manuscript.
It appears this module provides only slight modification, a short-term learning module based on self-attention, along with a transformer. The authors should clarify the full set of modifications made to the transformer structure to adapt it for gesture recognition tasks.

5. 'We pre-train on GRABMyo for 20 epochs using a fixed learning rate of 1e-4 for the backbone.' (line 307-308).
The manuscript states that GRABMyo was used for pretraining. However, the code indicates that pretraining was conducted on Ninapro DB2. Please see evidence: (1) https://anonymous.4open.science/r/short_term_semg/pretrain.py,  code line 21:  train_loader, val_loader = get_dataloader_db2(dataCfg, "EMG_CSV/S1_E1_A1/"); (2) https://anonymous.4open.science/r/short_term_semg/cfg/db2.yaml;
(3) https://anonymous.4open.science/r/short_term_semg/dataloaders/Ninapro.py, code line 67, function get_dataloader_db2(cfg, path_s,exercise), which returns data from Ninapro.
The descriptions in the manuscript and code appear contradictory and require clarification. Furthremore, the manuscript does not discuss which dataset was used for fine-tuning. Based on the code in  https://anonymous.4open.science/r/short_term_semg/finetuning.py, fine-tuning seems to employ two datasets, namely "hospital" and Ninapro DB2.

6. Unclear Details on Dataset and Evaluation Protocol.
Clearer descriptions and justifications are requied:
(1) Whether this work follows the protocol proposed by the original paper that published the dataset. If not, please provide justification, as the dataset’s original paper [2] appears to use a different evaluation protocol;
(2) Whether results for comparison methods directly reported from recent works, or re-implemented by the authors (e.g. Table 1);
(3) For works used in comparison (e.g., [3]) that employed the same dataset, was the same evaluation protocol followed? If not, please provide justification;
(4) Specify whether the experiments were participant-dependent or participant-independent;
(5) Clearly state the sizes of training, validation, and test sets.


7. Fairness of Comparison.
(1) Pretraining: Most methods used for comparison (except [2]) were not pretrained on EMG data, while the proposed method employs comprehensive pretraining and fine-tuning stages. This may lead to an unfair comparison.
(2) Ablation Study: The contributions of proposed innovations appear marginal based on Table 2 and Table 3. Using focal loss seems to lead significantly better performance than cross-entropy. For fair comparison, consistency across methods (focal loss vs. cross-entropy) is recommended.


8. Generalizability.
The generalizability of the proposed model remains uncertain as only one dataset was used for evaluation. In [3], a comparison method used in this work, has been evaluated on multiple EMG datasets. Additional dataset(s) should be used to evaluate the generalizability of the proposed framework.



Minor concerns :
1. 'STEM generalizes across different gesture recognition tasks' (line 029-030).
Since the experiments are conducted using only one dataset, it would be more appropriate to state that the model "generalizes across different gesture recognition tasks within the GRABMyo dataset." This would avoid any implication of broader cross-dataset generalization.

2. Use of Linear Projection.
Figure 2(b) shows that linear projection is applied in the long-term enhanced module but not in the short-term enhanced module. The rationale behind this selective application of linear projection requires further clarification.

3. Naming of 'Short-Term Enhanced Transformer' for the Proposed Method.
The proposed network integrates both short-term and long-term enhancement modules, yet is named "Short-Term Enhanced." This naming could be misleading, suggesting the network is optimized solely for short-term enhancement.




[1] Kim, Changjin, Tae Hyun Kim, and Sungyong Baik. "LAN: Learning to Adapt Noise for Image Denoising." Proceedings of the IEEE/CVF Conference on Computer Vision and Pattern Recognition. 2024.

[2] Pradhan, Ashirbad, Jiayuan He, and Ning Jiang. "Multi-day dataset of forearm and wrist electromyogram for hand gesture recognition and biometrics." Scientific data 9.1 (2022): 733.

[3] Zhang, Wenli, et al. "LST-EMG-Net: Long short-term transformer feature fusion network for sEMG gesture recognition." Frontiers in Neurorobotics 17 (2023): 1127338.

**Questions:**

See comments in the 'Weakness' section above. The current version of the manuscript is not satisfactory, but I am willing to raise my score if the authors can address any of my major concerns.

---

### Comment · Area_Chair_MM92 · 2024-11-21
**Rebuttal**

Dear Reviewers,

I encourage you to review the rebuttal and reach out to the authors with any additional questions or requests for clarification.

Best,\
AC

---

### Meta-Review · Area_Chair_MM92 · 2024-12-15

**Metareview:**

The paper proposes a method for gesture recognition from sEMG. The work tackles an important area and shows strong results. The paper is easy to follow and understand. The shortcomings of the paper initially raised by the reviewers include the lack of novelty in individual components of the method, limited dataset scope, lack of diverse baselines and comparisons with state-of-the-art methods, missing references, and fairness in evaluation protocols.

**Additional Comments On Reviewer Discussion:**

The reviewers appear divided in their assessments, with scores of 3, 5, 6, 6, 8. I have carefully read the paper, the reviews, and the rebuttal. While I disagree with some of the critiques brought up by Reviewer 5Rea (particularly regarding the validity of the replicated results and the importance of the completeness of the provided code), I do have some reservations about the paper. Specifically, I agree with the assessments of Reviewers bTGn as well as 5Rea that the paper falls short in the following areas:

1. While the paper explains the methodology ("what") clearly, it does not explicitly address "why" the proposed mechanisms are particularly well-suited for sEMG. This leaves an important gap in the paper's justification of its approach. As an extension to this, I do find the general contributions/novelty of the work to be more on the limited side.

2. The experimental evaluation is relatively narrow in scope. Since the paper is highly empirical and does not provide significant theoretical contributions, the expectation is that the experiments would be comprehensive; which is not the case. In particular, the paper's primary claim revolves around robustness to noise, yet the experiments only consider synthetic noise, i.e., Gaussian noise and signal loss. This, I believe is a significant shortcoming of the paper. The paper should have conducted more diverse and in-depth analyses, particularly with respect to real-world noise such as motion artifacts, which are commonly encountered in sEMG systems.

While some of the initial concerns of the reviewers with the paper were alleviated during the rebuttal (e.g., Reviewer 5Rea raised their score from 1 to 3), the above two weaknesses remained, which were deemed significant by the reviewers.

---

### Decision · Program_Chairs · 2025-01-22

Reject